# Antioxidant and Chemopreventive Activity of Protein Hydrolysates from Raw and Germinated Flour of Legumes with Commercial Interest in Colorectal Cancer

**DOI:** 10.3390/antiox11122421

**Published:** 2022-12-08

**Authors:** Marco Fuel, Cristina Mesas, Rosario Martínez, Raúl Ortiz, Francisco Quiñonero, Francisco Bermúdez, Natalia Gutiérrez, Ana M. Torres, Garyfallia Kapravelou, Aída Lozano, Gloria Perazzoli, Jose Prados, Jesús M. Porres, Consolación Melguizo

**Affiliations:** 1Institute of Biopathology and Regenerative Medicine (IBIMER), Center of Biomedical Research (CIBM), University of Granada, 18100 Granada, Spain; 2Department of Anatomy and Embryology, Faculty of Medicine, University of Granada, 18071 Granada, Spain; 3Instituto Biosanitario de Granada (ibs.GRANADA), 18014 Granada, Spain; 4Cellbitec S.L., N.I.F. B04847216, Scientific Headquarters of the Almería Technology Park, Universidad de Almería, 04128 La Cañada, Spain; 5Department of Physiology, Institute of Nutrition and Food Technology (INyTA), Biomedical Research Center (CIBM), Universidad de Granada, 18100 Granada, Spain; 6IFAPA Centro Alameda del Obispo, Área de Genómica y Biotecnología, Apdo 3092, 14080 Córdoba, Spain

**Keywords:** Fabaceae, antioxidant, chemoprevention, functional extracts, colon cancer

## Abstract

Legumes are a highly nutritious source of plant protein, fiber, minerals and vitamins. However, they also contain several bioactive compounds with significant potential benefits for human health. The objectives of this study were to evaluate the antioxidant, antitumor and chemopreventive activity of functional extracts from legumes using raw and germinated flours of six legume species of commercial interest. The methodology carried out consisted on the development of protein hydrolysates, assessment of their antioxidant capacity and *in vitro* tests on T84, HCT15 and SW480 colorectal cancer (CRC) cell lines. Our results showed a high antitumor activity of protein hydrolysate from *M. sativa*. Likewise, when combined with 5-Fluorouracile (5-Fu), there was a synergistic effect using extract concentrations from 50 to 175 µg/mL and 5-Fu concentrations from 1.5 to 5 µM. Similarly, the induction effect on detoxifying enzymes by the extracts of *M. sativa*, germinated *V. faba* Baraca × LVzt1 and *V. narbonensis*, which produced a higher induction rate than the positive control sulforaphane (10 µM), should be highlighted. Therefore, incorporating these enzymes into the diet could provide nutritional effects, as well as play an effective role in cancer chemoprevention and therapy.

## 1. Introduction

Risk factors such as a family history of colorectal cancer (CRC), inflammatory bowel disease, smoking, high consumption of red meat, age, obesity and alcoholism, among others, increase the incidence and mortality from colorectal cancer each year [1]. In fact, CRC is the third most common tumor type worldwide, representing the 10% of all tumors [2]. Although mortality rates are likely to decline in the coming years due early detection and new treatments, CRC recurrences and metastases represent a serious health problem [3]. Currently, the treatment of advanced CRC is based on chemotherapy (5-fluorouracil (5-FU), oxiplatin and capecitabine, among others) and monoclonal antibodies. However, their limited clinical efficacy, the development of drug resistance and their toxic side effects are serious limitations in CRC treatment. In this context, it has been demonstrated that several natural active compounds from plants show activity against CRC by modulating signaling pathways, regulating gene expression and controlling apoptosis [4,5]. In addition, the association between natural bioactive components and chemotherapeutic drugs has shown favorable results in CRC treatment related to the control of metabolic pathways mechanisms and through a synergistic effect that can reduce the development of resistance and adverse reactions to chemotherapy drugs [6,7].

Legumes are important sources of nutrients in human and animal diets and have been used throughout history in multiple countries. They confer significant benefits to human health due to their high content of proteins, dietary fiber, unsaturated fatty acids, vitamins, iron and zinc [8]. In addition to their nutritional value, legumes have recently gained interest because their frequent consumption supports health and disease mitigation through their nutritional profile and bioactive compounds [9]. Legumes help the homeostatic control of lipids and may potentially prevent cardiometabolic risks [10] or diseases, such as ischemic heart disease, and type 2 diabetes mellitus [11]. In addition, they modulate the gut microbiome, improve glycemic control and reduce cholesterol absorption. Furthermore, they exhibit interesting benefits on weight control by acting on the hormones that regulate appetite and satiety [12]. Resistant starch, phytate, polyphenols, oligosaccharides and saponins are among the most important bioactive compounds present in legumes, with many of them acting as antioxidants and antiproliferative agents [9]. Moreover, functional extracts derived from different legume species have shown high antioxidant and anti-proliferative activity against different cancer cells [13,14].

Legume protein hydrolysates exhibit some properties that are of vital importance for nutrition and disease control. For example, the peptides present in protein hydrolysates show strong antioxidant activity since certain amino acids act as chelating agents of metals or hydrogen/electron donating agents, thus interfering with the formation of free radicals. Some examples of these amino acids are tryptophan, phenylalanine and histidine, where the donation of hydrogen bonded to the nitrogen of its indole ring eliminates free radical development. Similarly, certain active peptides from legumes are able to significantly act in blood pressure control and modulate immune, neurochemical and brain function in humans. These characteristics play a key role in the fight against metabolic and chronic diseases such as obesity, type II diabetes, immunosuppression, neurodegenerative diseases, cancer and other age-associated disorders resulting from oxidative stress. In this regard, research on protein hydrolysates and their bioactive peptides has attracted attention as they may be safer alternatives for therapeutic applications [15].

Despite current knowledge about the legume bioactive compounds, new studies are still needed to focus on their pharmacological antitumor activity and their action mechanisms. In our study, we obtained protein hydrolysates from different legume seeds to study their antioxidant, antitumor and chemopreventive activity against CRC cell lines. Our results demonstrated high antioxidant and chemopreventive activity against this type of tumor cells, indicating that they can be as a source of new compounds to improve available antitumor therapies. Since CRC is a multifactorial disease, these natural extracts or their bioactive components could be applied as an adjuvant therapy to enhance the antitumor effect of the classic drugs used in this pathology and to reduce their side effects.

## 2. Materials and Methods

### 2.1. Chemicals and Reagents

Hydrogen peroxide solution, 5-Fluorouracil, trhizome^®^ base, malondialdehyde (MDA), thiobarbituric acid (TBA), 3-(4,5-Dimethyl-2-Thiazolyl)-2,5-Diphenyl tetrazolium bromide (MTT), gallic acid, glutathione (GSH, reduced form), 1-chloro-2,4-dinitrobenzene (CDNB), β-nicotinamide adenine dinucleotide (NAD, reduced disodium salt hydrate), flavin adenine dinucleotide disodium (FAD, salt hydrate), 2.6-dichloroindophenol (2.6-DCIP, sodium salt hydrate) and DL-Sulforaphane (SFN) were purchased from Sigma-Aldrich, Madrid, Spain.

### 2.2. Plant Material and Germination Conditions

Seed flours from six legume species of commercial interest were used in the study: faba bean (*Vicia faba*), narbonne vetch (*Vicia narbonensis*), bitter vetch (*Vicia ervilia*), common vetch (*Vicia sativa*), yellow lupin (*Lupinus Luteus*) and alfalfa (*Medicago sativa*). Faba bean lines and varieties were classified in relation to the low or high presence of tannins and vicine/convicine (var. Chipen, var. Aldaba, line Cana, line Alameda × LVzt2 and line Baraca × LVzt1) (Table 1). All legume seeds were provided by the Instituto Andaluz de Investigación y Formación Agraria, Pesquera, Alimentaria y de la Producción Ecológica (IFAPA) (Córdoba, Spain) and CELLBITEC S.L (Almería, Spain). The seeds grown at IFAPA, were harvested in 2018 and 2019 and stored at room temperature in a dry environment until analysis. For germination, the methodology described by Urbano et al. [16] and Kapravelou et al. [17] was implemented with small modifications. In total, 100 g of each faba bean seed type was cleaned to remove any impurities, sterilized in sodium hypochlorite for 3 min and then washed and soaked in sterile distilled water for 8h. The hydrated seeds were sown on moistened germination paper at 30 °C, in the dark, for four days. Uniformly germinated seeds were grinded using a Mixer Mill MM 400 (Retsch, Biometa Tecnologia y Sistemas, S.A., Asturias, Spain), lyophilized and grinded again to obtain a fine ground flour, while raw seeds were only milled once. Germinated and raw flours were stored in a plastic container with a lid until analysis. Legume flours were used to obtain protein hydrolysates.

### 2.3. Preparation of Protein Hydrolysate

Legume protein hydrolysates were prepared by a simultaneous process of alkaline water extraction and hydrolysis with proteases as described by Kapravelou et al. [18]. In total, 25 g flour and 0.25 g sodium sulfite were resuspended in 100 mL distilled water. The pH was adjusted to 8.8 with 3N KOH and the temperature was set to 33 °C, with a stirring speed of 300 rpm for 30 min. The legume flour was then centrifuged at 3000 rpm for 5 min. The supernatant was saved, and the pellet was resuspended in 50 mL of distilled water, repeating the above process. The two supernatants obtained were mixed, and sufficient amounts of 100 mM CaCl_2_ and 100 mM MgSO_4_ solutions were added to reach a final 1 mM concentration of both. The mixture was incubated at 47 °C for 20 min under continuous agitation. Subsequently, enzymatic digestion started with the addition of *Bacillus licheniformis* protease (0.3 AU/g protein) at 47 °C and pH 8.8 for 30 min. Then, protease from *Aspergillus oryzae* (100 AU/g protein) was added under the same conditions. Finally, the samples were frozen at −80 °C and lyophilized for 48 h. The protein hydrolysates obtained were solubilized in type I water and heated at 95 °C for 10 min before being added to the cell cultures to inactivate the proteolytic enzymes.

### 2.4. Characterization of Antioxidant Capacity

#### 2.4.1. Quantification of Total Polyphenols

The total polyphenol content of protein hydrolysates from legume seeds was assessed by a modified Folin–Ciocalteu colorimetric assay [18]. A gallic acid standard curve (0–500 µg/mL) was used to determine the concentration, and the results are expressed as µg gallic acid equivalents (GAE) per mg of sample.

#### 2.4.2. ABTS Radical Scavenging Assay

The ABTS assay of total antioxidant capacity is based on the methodology of Miller et al. [19], which uses 2,2′-Azino-bis(3-ethylbenzothiazoline-6-sulfonic acid) (ABTS) as a free radical generator. In total, 6 µL of protein hydrolysate, or a standard solution of gallic acid (0–60 mg/L), was mixed with 294 µL of ABTS and incubated for 3 min. The optical density of the samples was then measured at 620 nm (Mul-tiskan™ FC, Microplate Photometer, Thermo Fisher Scientific, Waltham, MA, USA). The blank was made with 6 µL of water and 294 µL of ABTS. The results are expressed as µg of gallic acid equivalents (GAE) per mg of sample.

### 2.5. Cell Culture

The T84, SW480, HT-29 and HCT-15 (resistant to chemotherapy) human CRC cells were obtained from American Type Culture (ATCC) (Manassas, VA, USA). The non-tumor CCD-18 colon cell line (human colon epithelial cell line) was used as control and was provided by the Scientific Instrumentation Center (CIC, Granada University, Granada, Spain). All cell lines were grown in Dulbecco’s Modified Eagle’s Medium (DMEM) (Sigma-Aldrich, St. Lous, MO, USA), supplemented with 10% heat-inactivated fetal bovine serum (FBS) (Thermo Fisher Scientific, Waltham, MA, USA) and antibiotics (gentamicin/amphotericin – B + penicillin/streptomycin) (Sigma Aldrich, Madrid, Spain) at 1%, and maintained in an incubator at 37 °C with a 5% CO_2_ humidified atmosphere.

### 2.6. In Vitro Antioxidant Capacity

To assess the in vitro antioxidant capacity of legume protein hydrolysates, the HT-29 CRC cell line was used. A total of 5 × 10^4^ cells/well were seeded in 96-well plates. After 24 h, the culture medium was replaced with serum-free medium and incubated for 24 h. The protein hydrolysates were then added at non-cytotoxic doses and incubated for 24 h. Subsequently, the culture medium was discarded, and the oxidizing agent paraquat was added at concentrations of 25 mM and incubated for 6 h. Then, the medium was replaced by serum-free medium for 12 h. The cell viability was determined by an MTT assay to determine the relative proliferation (%PR) of the treated cells. The results of this test are expressed as Antioxidant Activity Units (UAA), which is defined as the value of 10 percentage units (10%) recovery of cell viability with respect to the corresponding control treated with paraquat.

### 2.7. Cell Viability Assay

To investigate the effect of protein hydrolysates on CRC cell proliferation, T-84 (4 × 10^3^ cells/well), SW480 (5 × 10^3^ cells/well), HCT-15 (5 × 10^3^ cells/well) and CCD18 (4 × 10^3^ cells/well) were seeded in 48-well plates. After 24 h, cell cultures were exposed to the protein hydrolysates dissolved in DMEM without any additional solvent. Then, cell cultures were exposed to increasing concentrations of protein hydrolysates for 72 h. In addition to testing them as a monotherapy, we also tested a combined therapy of protein hydrolysates with 5-Fu at different concentrations. After treatment exposure (72 h), cells were fixed with 10% trichloroacetic acid (TCA) (20 min at 4 °C). Once dried, the plates were stained with 0.4% sulforhodamine B (SRB) in 1% acetic acid (20 min, in agitation). After three washes with 1% acetic acid, SRB was solubilized with Trizma^®^ (10 mM, pH 10.5). Finally, the optical density (OD) at 492 nm was measured in a spectrophotometer EX-Thermo Multiskan (Thermofisher, Waltham, Massachusetts, USA). Cell survival (%) was calculated according to the following equation: Cell survival (%) = [(Treated cells OD − blank)/(Control OD − blank)] × 100. In addition, the half-maximal Inhibitory Concentration (IC_50_) was calculated (GraphPad Prism 6 Software, La Jolla, CA, USA). For the combination effect, the combination index (CI) was calculated using Compusyn software [20], where a CI > 1 indicates antagonism, a CI level <1 indicates synergy and a CI level equal to 1 indicates additivity.

### 2.8. Wound-Healing Assay

To determine the tumor cell migration capacity of cell lines and, therefore, their invasiveness and ability to generate metastases. T-84 cells (3 × 10^5^) were seeded in 12-well plates and grown to 100% confluence in standard culture conditions. Once confluence was reached, a “wound” was manually performed with a 100 µL sterile pipette tip following [21], and the medium was substituted for serum-free DMEM. Cells were immediately exposed to the protein hydrolysates (non-cytotoxic dose IC15) for 72 h. Images were obtained at different times (0, 24, 48 and 72 h) with an inverted light microscope Olympus CKX41 (Olimpus Corporation, Tokyo Japan)) to observe cell migration in comparison to the control (cells without treatment). To evaluate the effect of the protein hydrolysates, the percentage of migration was calculated by measuring the area free of tumor cells at different times (Image J software 1.53e) (https://imagej.nih.gov) (accessed on 12 September 2021).

### 2.9. Detoxifying Enzyme Induction

#### 2.9.1. Treatment and Purification of the Cytosolic Fraction

HT29 colon adenocarcinoma cells were seeded in T25 culture flask at a concentration of 1 × 10^6^ in supplemented DMEM and incubated for 24 h. Then, cells were exposed to the protein hydrolysates of the seeds for 48h using non-cytotoxic doses. Sulforaphane was used as a positive control at two concentrations (5 µM and 10 µM). After this incubation period, the medium was removed, and the cells were washed with PBS and trypsinized. Trypsin activity was neutralized with twice the amount of supplemented DMEM medium, and cells were transferred to 1.5 mL Eppendorf tubes and centrifuged at 10,000× *g*, 4 °C for 5 min. The supernatant was discarded, and the pellet was re-suspended into 500 µL of PBS and centrifuged under the same conditions. PBS was discarded, and the cells were re-suspended into 500 μL of 25mM Tris-HCl, pH-6.4. The cells were then lysed by sonication to 40% frequency for 10 s on ice and centrifuged at the same conditions. The enzyme activity was determined in the cytosolic supernatant.

#### 2.9.2. Glutathione S-Transferase (GST) Assay

The inactivation of genotoxic and cytotoxic compounds occurs through the action of the enzyme GST, which catalyzes the nucleophilic addition of glutathione to an electrophilic center that is part of the xenobiotics. Although the enzyme alone cannot reach its maximum functional capacity, a variety of compounds can induce its activity and increase it to provide effective protection against carcinogenesis. The GST assay was measured by observing the conjugation of 1-chloro-2,4-dinitrobenzene (CDNB) (molar extinction 9.6 mM^−1^ cm^−1^) with reduced glutathione (GSH). The reaction mix contained 980 μL of 100 mM phosphate buffer (pH 6.5), 10 μL of 100 mM reduced glutathione (GSH) and 10 μL of 100 mM CDNB. In total, 100 μL of each sample (cytosolic supernatant) and PBS (for the blank) were added to a cuvette containing 1 mL of the reaction mix, and the absorbance was measured at 340 nm each minute for 5 min. The GST activity was calculated as the increase in absorbance per min per mg total protein of the sample.

#### 2.9.3. NAD(P)H: Quinone Oxyidoreductase (QR) Assay

To avoid the toxicity of quinone and quinoneimine-type compounds, the cytosolic flavoprotein enzyme quinone oxidoreductase is responsible for reducing these compounds to their corresponding hydroquinones using NADH and NADPH as donors, thus avoiding the generation of semiquinone intermediates since these compounds have a high tendency to react with oxygen and convert to superoxide. The QR assay was measured by observing the reduction of 2.6-dichloroindophenol (2.6-DCPIP) (molar extinction 0.0205 μM^−1^ cm^−11^) by QR. The reaction mix contained 881.5 μL of 25 mM Tris-HCl (pH-6.5), 10 μL of 20 mM NADH, 5 μL of 10 μM FAD, 60 μL of BSA (1mg/mL), 2.5 μL of Tween (20%) and 16 μL of 5 mM DCPIP. In total, 25 µL of each sample (cytosolic supernatant) and Tris-HCl (for the blank) were added to a cuvette containing 1 mL of the reaction mix, and the absorbance was measured at 600 nm each minute for 5 min. The QR activity was calculated as the decrease in absorbance per min per mg total protein of the sample.

### 2.10. Statistical Analysis

Statistical analysis was performed by IBM SPSS Statistics 26.0 and GraphPad Prism 8. All data are expressed as the mean value with standard deviation (SD), and all experiments were performed in triplicate. The results obtained from the antioxidant tests were analyzed by a one-way ANOVA. The Tukey-Kramer test was used to detect the differences between treatment means. The analyses were performed with Statistical Package for Social Sciences (IBM SPSS for Windows, version 22.0, Armonk, NY, USA), and the level of significance was set at *p* < 0.01. After the homogeneity test of variance, Student’s *t*-test was performed when comparing the difference between groups with equal variance, while an F-test was performed for groups with uneven variance. Significant values were denoted by (*) *p* < 0.05, significant; (**) *p* < 0.01, highly significant; and (***) *p* < 0.001, very highly significant.

## 3. Results

### 3.1. Analysis of Yield and Antioxidant Activity

The yield and antioxidant activity of the protein hydrolysates from *V. faba* seeds (raw and germinated flour) are shown in Table 2. Protein hydrolysates from germinated *V. faba* beans exhibited higher extraction yields (*p* < 0.05) compared to raw beans, with the germinated Cana line standing out (510.3 mg/g of flour). Furthermore, the germinated broad beans showed the highest total polyphenol content. Regarding the ABTS test, the raw beans showed higher activity compared to the germinated ones, and the highest value corresponded to the raw var. Chipen (5.26 µg GAE/mg extract).

As for the other legume species studied (Table 3), although the highest yield was obtained in *L. luteus* (553.6 mg/g flour) compared to the rest of legume crops, *M. sativa* showed the highest result in the ABTS test (12.18 µg GAE/mg PH) as well as the highest result in total polyphenols (91.8 µg GAE/mg PH).

### 3.2. Antioxidant Activity of Cultured Cells

The antioxidant capacity of legume protein hydrolysates tested in a HT29 cell line subjected to oxidative stress using a free radical generator (paraquat) is presented in Table 4 and Table 5. The protein hydrolysates from the different *V. faba* lines and varieties showed a great antioxidant capacity against oxidative stress, especially the extracts of the germinated Baraca × LVzt1 line. In general, the raw seeds showed higher antioxidant activity in vitro than the germinated seeds. Compared to the other legume species, *M. sativa* exhibited the highest antioxidant capacity, followed by the three species of the genus *Vicia*, among which *V. sativa* showed the highest antioxidant capacity. However, in case of *L. ervilia*, its antioxidant capacity was moderate.

### 3.3. Antiproliferative Activity

The study of cell viability in the CRC lines showed that the protein hydrolysate of the germinated Baraca × LVzt1 line induced a significant antitumor activity in the T84 (IC_50_ value of 360.8 µg/mL) and SW480 CRC cells (397.3 µg/mL). The rest of faba bean lines or varieties did not present relevant antitumor activity (Table 6) (Appendix A).

Among the rest of the analyzed legumes, the *M. sativa* protein hydrolysate demonstrated the highest antiproliferative activity against the T84, SW480 and resistant HC-T15 cells (IC_50_ 269.2, 328.2 and 452.8 µg/mL, respectively). This extract also modulated the proliferative capacity of CCD18, although at very high concentrations (IC_50_ 710.4 µg/mL). Similarly, the protein hydrolysate from *V. narbonensis* showed significant antitumor activity in the T-84 (IC_50_ 328.8 µg/mL) and SW4480 cells (405.5 µg/mL) (Table 6).

### 3.4. Synergistic Effect between 5-FU and Legume Extracts

The synergistic effect of protein hydrolysates with the drug 5-FU was analyzed in the T-84 cell line. The protein hydrolysate of *M. sativa* demonstrated a significant synergistic effect with the agent 5-FU compared to the treatment with 5-FU alone (IC < 1) (Figure 1).

### 3.5. Cell Migration Analysis

To evaluate the influence of protein hydrolysates on the migration capacity of tumor cells, a cell wound migration assay was performed in the T84 line with the protein hydrolysates that showed antitumor activity. The protein hydrolysates of the germinated *V. faba* line Baraca × LVzt1, *V. narbonensis* and *M. sativa* demonstrated a significant decrease in tumor cell migration compared to the control (untreated) cells. The inhibitory effect was observed as early as 24 h and was more pronounced at 72 h (*p* < 0.05) (Figure 2).

### 3.6. Legume Extracts and Detoxifying Enzyme Activity

The detoxifying enzyme activity was analyzed in the HT-29 cells after legume extract exposure. As shown in Table 7, the activity of drug-metabolizing enzymes GST and QR was induced by protein hydrolysates from *M. sativa*, germinated *V. faba* line Baraca × LVzt1 and *V. narbonensis* compared to the control (sulforaphane). It is noteworthy that the protein hydrolysate extract from *M. sativa* induced the highest activity of these enzymes compared to the other species and the positive control (Sulforaphane) (Table 7).

## 4. Discussion

The treatment of CRC has progressed significantly in recent years due to the understanding of the molecular mechanisms involved in its pathogenesis, the design of new drugs such as monoclonal antibodies and improved chemotherapeutic as agents and the use of cell and gene therapies [22]. In some cases, however, the complexity of the disease and therapeutic limitations produce adverse effects such as toxicity in healthy tissues and the development of chemoresistance. For this reason, it is necessary to search for new bioactive agents in plants with metabolites that have shown safety, efficacy and synergism with therapeutic agents [23]. On the other hand, it has been shown that the intestinal microbiome influences the development of CRC. In fact, certain risk factors for CRC also affect the composition of the intestinal microbiota (obesity, physical inactivity, red meat intake, etc.), whereas dysbiosis can promote chronic inflammatory conditions and the production of toxic and carcinogenic metabolites that lead to dysplasia and neoplasia. Furthermore, an increased population of *Fusubacterium nucleatum*, enterotoxigenic *Bacteroides fragilis* and some strains of enterotoxigenic *E. coli* have been related to low antitumor immune response, systemic inflammation mediated by LPS (lipopolysaccharide endotoxin) and depletion of SCFA (short chain fatty acid), among other effects [24].

Plant-derived bioactive compounds, including certain phenolic derivatives and bioactive peptides, have been shown to modulate the composition of the intestinal microbiota (inhibiting the population of pathogens and promoting the growth of beneficial bacteria). They also prevent the production of toxic compounds such as LPS, hydrogen sulfide and indole and increase the production of beneficial metabolites such as SCFA and other bioactive compounds that target multiple pathways and tissues, resulting in improved intestinal health, glycemic and lipid control and inflation [25,26]. Regarding the relationship between the bioaccesibility and bioactivity of plant bioactive compounds such as polyphenols and the action of the human microbiota, it is important to consider that only a proportion of the daily amount of bioactive compounds ingested is absorbed or metabolized by certain bacterial strains. Therefore, new strategies have been developed to increase their bioavailability using probiotics (*Saccharomyces cerevisiae*, *Saccharomyces boulardii* or *Lactobacillus* (L.) *plantarum*) and prebiotics, or by reducing the yield of biotransformations that limit the expression of bioactivity. The beneficial effects of functional compounds can be enhanced by modulating the microbiota to transform these substrates, acting as a substrate for the enzymatic apparatus of the microbiota or as a carbon source, as well as modulating the population of microorganisms due to the antimicrobial effect that many polyphenols have, thus avoiding dysbiosis. In addition, another advantage is their easy administration in the form of biomass enriched with functional compounds that improve the assimilation of the active principle in the human colon, increase resistance to oxidative stress resulting from dysbiosis and maintain a greater viability, Thus, functional compounds can exert an effect for a longer time, which is an essential property in the process of the modulation of the intestinal microbiota [27,28].

*Vicia faba* is the fourth most important legume crop, with over 5.7 million tons being harvested globally in 2020 [29]. Recently, the interest in this species has grown due to its nutritional characteristics and health-benefitting properties [30,31]. In addition to a variety of bioactive compounds with demonstrated antioxidant activity, such as total phenolics, and flavonoids [32], faba bean contains two main antinutritional factors (tannins and vicine-convicine) that decrease the digestibility and biological value of the protein in animal feeding. Vicine-convicine also causes favism, a severe form of hemolytic anemia, in humans who have an X chromosome-inherited glucose-6-phosphate dehydrogenase (G6PD) deficiency [33]. In faba beans, vicine is found in concentrations between 0.44 and 0.82%, and convicine is found in concentrations between 0.13 and 0.64%. Both compounds are hydrolyzed in the digestive tract to the highly reactive free radicals divicine and isouramyl that cause hemolytic anemia [34]. For this reason, different faba bean lines varying in tannin and vicine-convicine content were used in the analyses. The functional extracts showed high antioxidant activity and high polyphenolic content, which is in agreement with the results of the authors of [35], who analyzed the pods of seven varieties of *Vicia faba* species, finding a high polyphenolic content and antioxidant activity. In our study, no correlation was observed between antioxidant activity and the concentration of tannins and vicine-convicine, with the exception of the ABTS assay.

In relation to *in vitro* antioxidant activity, protein hydrolysates from seeds with a higher tannin concentration showed a greater protective action against the oxidative stress generator paraquat. These results are similar to those obtained in the study of Oomah et al. [36], who found a correlation between tannin content and total polyphenols and reported the presence of a higher antioxidant activity in beans with higher tannin content. The same result was obtained when analyzing seeds of the legume *Vigna aconitifolia*, which showed high antioxidant, antidiabetic and antihypertensive capacity due to their tannin, catechin and gallic acid content [37].

Regarding the antitumor activity, the protein hydrolysate of germinated Baraca × LVzt1 line inhibited the proliferation of T84 and SW480 cell lines. This could be explained by the fact that protein hydrolysates provide bioactive peptides that increase the functional and antioxidant properties of bean proteins [38]. In addition, a possible molecule responsible for the antiproliferative activity called Soyasaponin I was found, which has been reported in some varieties of beans and other legumes [39]. Soyasaponin I is also able to inhibit the tumor growth and metastasis in cancer cells [40], which would explain this effect in our study.

In relation to the group of legumes with high protein content, they also presented high yields, polyphenol content and antioxidant activity, with *Medicago sativa* and *Vicia sativa* standing out. Our results are in agreement with Liu et al. [41], who reported a high antioxidant capacity and moderate cytotoxic activity against Hela tumor cells by the chloroform-phase organic extract of the stems and leaves of *Vicia sativa L*. The authors attributed their results to the presence of phenolic compounds, flavones, curcumin and triterpenoids. The antiproliferative activity of *Medicago sativa* protein hydrolysate is associated with the presence of bioactive peptides rich in cysteine [42] such as *Medicago Sativa* Defensin I peptide (MsDef1), which has shown antitumor activity in MDR cancer cells [43]. Regarding the antiproliferative activity and effects on tumor migration by the protein hydrolysate of *Vicia narbonensis*, no similar studies have been reported so far. However, bioactive compounds such as flavonoid glycosides of the kaempferol and quercetin glycosides type have been identified [44] which have shown antitumor activity [45].

The correlation between antioxidant and antitumor activity could be related to the chemical structure of the bioactive compounds, as well as the presence of certain functional groups such as hydroxyls. It has been reported that in the case of trihydroxyflavone, the ortho-dihydroxy structural fragment in ring B is responsible for the anticancer and antioxidant activity in A549 and U87 cells, whereas it also correlates directly with DPPH radical scavenging activity [46].

Another property to take into consideration in the treatment of cancer is the modulation of phase II detoxifying enzymes such as glutathione-s-transferase and quinone oxidoreductase, which protect cells from xenobiotic agents, oxidants and secondary metabolites that are toxic to cells. Therefore, their induction by natural agents represents a promising strategy for their prevention [47]. Thus, in our study, an induction by the functional extracts of the germinated *Vicia faba* line Baraca x LVzt1, *Vicia narbonensis* and *Medicago sativa* was evidenced. Our results are in agreement with other studies of other legume crops, such as lentils, peas or butter beans, revealing the ability to induce these enzymes [48].

## 5. Conclusions

Protein hydrolysates of traditional legume crops were evaluated for their *in vitro* antioxidant capacity, their effect on the proliferation and inhibition of migration in human CRC lines, as well as their role in the induction of phase II detoxification enzymes. All *Vicia faba* lines studied revealed similar antioxidant activities regardless of their tannin or vicine-convicine concentrations. For the remaining legume crops with higher protein content mostly used in animal feeding, a high antioxidant capacity was evidenced, with *Medicago sativa* standing out for its antitumor capacity. In addition to these results, the chemopreventive activity exhibited by some legumes could result in the development of functional food products with antioxidant and neoadjuvant potential for cancer therapy, which should be further explored in future research.

## Figures and Tables

**Figure 1 antioxidants-11-02421-f001:**
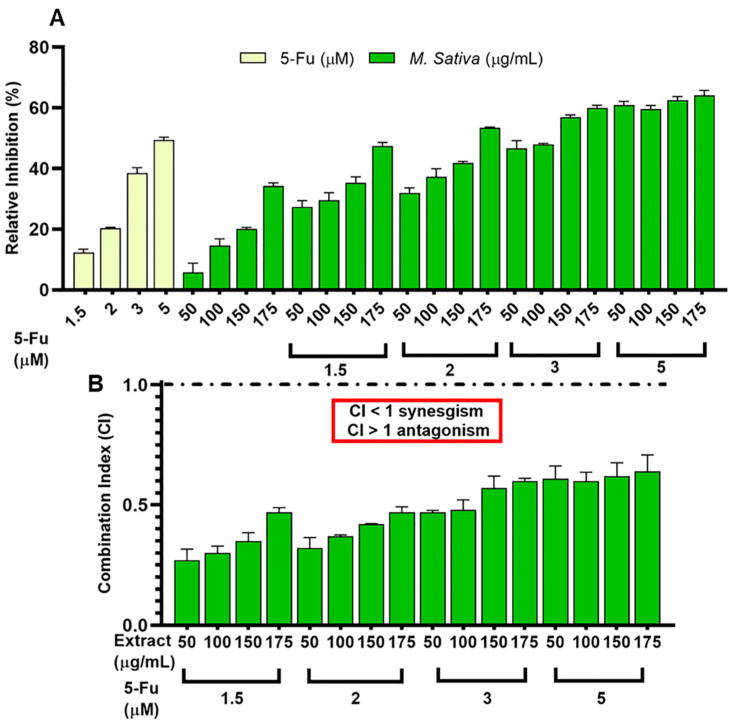
Antiproliferative effect of the combined treatment (protein hydrolysate from *M. sativa* and 5-Fu) on the T-84 cells. (**A**) *M. sativa* protein hydrolysate and 5-Fu. (**B**) Combination index (CI) values of *M. sativa* protein hydrolysate and 5-Fu are shown above the bars. CI ≤ 1 and >1 indicate synergism, addition and antagonism, respectively.

**Figure 2 antioxidants-11-02421-f002:**
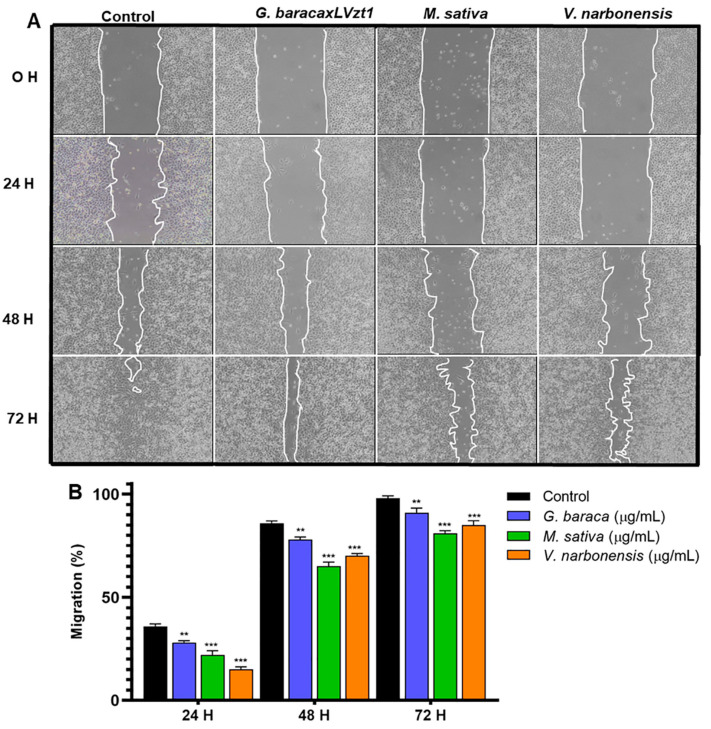
Analysis of the migration inhibitory capacity. (**A**) Representative images of the cell migration of the T84 line (0, 24, 48, 72 h) after exposure to a non-cytotoxic dose (IC15) of the protein hydrolysate from *V. faba* line Baraca × LVzt1 (germinated), *M. sativa* and *V. narbonensis*. (**B**) Graphical representation of the percentage of the migration of T84 tumor cells. Data are reported as mean ± SD with experiments performed in triplicate. Significant values are denoted by (**) *p* ≤ 0.01 highly significant, (***), *p* ≤ 0.001 very highly significant.

**Table 1 antioxidants-11-02421-t001:** Concentration of tannins and vicin and convicin in the different faba bean lines and varieties.

Concentration	Lines/Varieties	
High Tannins/High V-C	Chipen	Raw
		Germinated
Low Tannins/High V-C	Aldaba	Raw
		Germinated
	Cana	Raw Germinated
Low Tannins/Low V-C	Alameda × LVzt2	Raw
	Baraca × LVzt1	Germinated

**Table 2 antioxidants-11-02421-t002:** Quantification of the total polyphenols and antioxidant capacity in different *V. faba* varieties (raw and germinated flour).

Varieties of *V. faba*		Yield (mg/g Flour)	Total Polyphenols (µg GAE/mg PH)	ABTS (µg GAE/mg PH)
Chipen	Raw	384.5 ± 1.89 A	34.9 ± 0.07 A	5.26 ± 0.02 A
	Germinated	478.9 ± 0.61 B	23.5 ± 0.14 B	3.62 ± 0.04 B
Aldaba	Raw	427.3 ± 2.95 C	28.9 ± 0.33 C	3.76 ± 0.07 C
	Germinated	477.5 ± 0.87 B	22.5 ± 0.11 D	2.61 ± 0.03 D
Cana	Raw	413.3 ± 0.59 C	18.1 ± 0.02 E	1.02 ± 0.03 E
	Germinated	510.3 ± 1.40 D	20.8 ± 0.03 F	1.87 ± 0.03 F
Alameda × LVzt2	Raw	351.9 ± 2.72 E	18.7 ± 0.17 E	0.61 ± 0.01 G
Baraca × LVzt1	Germinated	455.2 ± 0.90 F	31.6 ± 0.02 G	0.91 ± 0.01 H

Data are reported as mean ± SD with experiments performed in triplicate. PH: protein hydrolysate. GAE: gallic acid equivalent. Means within the same column with different letters differ significantly (*p* < 0.01).

**Table 3 antioxidants-11-02421-t003:** Quantification of the total polyphenols and antioxidant capacity in different legume species (only raw flour).

Legumes	Yield (mg/g Flour)	Total Polyphenols (µg GAE/mg PH)	ABTS (µg GAE/mg PH)
*L. luteus*	553.6 ± 1.72 A	13.8 ± 0.12 A	2.90 ± 0.01 A
*M. sativa*	384.4 ± 0.64 B	91.8 ± 0.82 B	12.18 ± 0.02 B
*V. ervilia*	385.6 ± 0.80 B	27.6 ± 0.11 C	2.51 ± 0.02 C
*V. narbonensis*	262.1 ± 4.03 C	24.1 ± 0.02 D	5.21 ± 0.06 D
*V. sativa*	365.8 ± 1.07 D	26.4 ± 0.06 C	3.57 ± 0.01 E

Data are reported as mean ± SD with experiments performed in triplicate. PH: protein hydrolysates GAE: gallic acid equivalent. Means within the same column with different letters differ significantly (*p* < 0.01).

**Table 4 antioxidants-11-02421-t004:** Antioxidant capacity of the protein hydrolysates from different *V. faba* varieties (raw and germinated flour) tested in HT-29 cell line.

Varieties of *V. faba*		*In Vitro* Antioxidant Activity (mUAA/mg)
		Paraquat
Chipen	Raw	302.9 ± 1.77
	Germinated	219.3 ± 1.16
Aldaba	Raw	143.0 ± 1.76
	Germinated	130.3 ± 2.18
Cana	Raw	267.7 ± 1.73
	Germinated	309.1 ± 1.89
Alameda × LVzt2	Raw	283.8 ± 0.63
Baraca × LVzt1	Germinated	368.9 ± 1.91

Results are expressed as the mUAA/mg of the protein hydrolysate. An UAA is defined as the value of 10 percentage units of cell viability recovery with respect to the paraquat-treated control. Data are reported as mean ± SD with experiments performed in triplicate.

**Table 5 antioxidants-11-02421-t005:** Antioxidant capacity of protein hydrolysates from different legume species tested in the HT-29 cell line.

Legumes	*In Vitro* Antioxidant Activity (mUAA/mg)
	Paraquat
*L. luteus*	273.2 ± 1.42 A
*M. sativa*	616.2 ± 1.58 B
*V. ervilia*	236.7 ± 1.98 C
*V. narbonensis*	301.4 ± 1.51 D
*V. sativa*	437.2 ± 1.61 E

Results are expressed as the mUAA/mg of the protein hydrolysate. An UAA is defined as the value of 10 percentage units of cell viability recovery with respect to the paraquat-treated control. Data are reported as mean ± SD with experiments performed in triplicate.

**Table 6 antioxidants-11-02421-t006:** Antiproliferative activity (IC_50_) of the protein hydrolysates against the CRC cell lines.

	IC_50_ (µg/mL)			
	T84	HCT-15	SW480	CCD18
***V. faba* var. Chipen**				
*raw*	-	-	-	-
*germinated*	-	-	-	-
***V. faba* var. Aldaba**				
*raw*	-	-	-	-
*germinated*	-	-	-	-
** *V. faba* ** **var. Alameda × LVzt2**				
*raw*	-	-	-	-
***V. faba* var. Cana**				
*raw*	-	-	-	-
*germinated*	-	-	-	-
** *V. faba* ** **var. Baraca × LVzt1**				
*germinated*	360.8 ± 2.35	-	397.3 ± 1.06	-
** *L. luteus* **	-	-	-	-
*raw*				
** *V. narbonensis* **	328.8 ± 3.46	-	405.5 ± 1.03	879.5 ± 1.20
*raw*				
** *M. sativa* **	269.2 ± 0.32	452.8 ± 5.49	328.2 ± 1.68	710.4 ± 5.90
*raw*				
** *V. ervilia* **	954.9 ± 0.58	-	-	-
*raw*				
** *V. sativa* **	-	-	-	-
*raw*				

Data are reported as mean ± SD with experiments performed in triplicate. (-): Not defined.

**Table 7 antioxidants-11-02421-t007:** GST and QR induction activity on the HT-29 cells after treatment with protein hydrolysates.

		GST			QR		
	Concentration of PH	U/mL	U/mg	Induction Rate (Treated/Control)	U/mL	U/mg	Induction Rate (Treated/Control)
Control		72.9 ± 2.66	20.6 ± 0.76	1.00 ± 0.00	229.4 ± 1.72	591.8 ± 0.24	1.00 ± 0.00
*M. sativa*	25 µg/mL	220.9 ± 4.18	51.9 ± 1.29	2.53 ± 0.05 ***	6684.4 ± 3.91	1768.4 ± 0.51	2.99 ± 0.02 ***
*V. faba* line Baraca × LVzt1 *germinated*	25 µg/mL	179.1 ± 1.80	37.9 ± 0.38	1.84 ± 0.02 ***	6383.4 ± 2.61	1305.4 ± 0.34	2.21 ± 0.01 ***
*V. narbonensis*	25 µg/mL	197.9 ± 3.43	37.9 ± 0.66	1.84 ± 0.03 ***	5661.7 ± 4.19	1139.2 ± 0.64	1.92 ± 0.03 ***
Sulforaphane	5 µM	66.3 ± 1.49	22.3 ± 0.50	1.08 ± 0.02 *	5852.4 ± 1.20	1336.1 ± 0.14	2.28 ± 0.01 ***
Sulforaphane	10 µM	116.2 ± 3.87	26.5 ± 0.88	1.29 ± 0.04 ***	4947.9 ± 2.64	1660.4 ± 0.12	2.81 ± 0.01 ***

Induction results expressed as a mean of ratio of GST activity of treated *vs*. control samples. Significant values are denoted by (*) *p* < 0.05 significant; (***), *p* ≤ 0.001 very highly significant.

## Data Availability

No applicable.

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
