# Peer review of "Antioxidant and Chemopreventive Activity of Protein Hydrolysates from Raw and Germinated Flour of Legumes with Commercial Interest in Colorectal Cancer"

_antioxidants, 2022, doi:10.3390/antiox11122421_

Round 1

Reviewer 1 Report

The manuscript “Antioxidant and chemopreventive activity from legumes of commercial interest in colorectal cancer” by  Fuel et al. investigates the antioxidant and antiproliferative properties of ethanolic extracts and protein hydrolysates from raw and germinated legumes in colon cancer cells.

Although the authors present a current research topic, their experimental design is poor, the manuscript is noncoherent, and the presentation of results makes the interpretation of the experiments difficult. Those are the primary issues that the authors should consider:

-        Title. It should be revised to include “germination”, “extracts”, and “protein hydrolysates”. The authors are not investigating the effects of whole seeds but isolated compounds from them.

-        Abstract. The authors must rewrite it and ensure it follows a logical order (objective > methods > results > conclusions). Make sure you define abbreviations if used.

-        Lines 55-57. Rewrite.

-        Lines 60-64. Rewrite.

-        Lines 69-76. This paragraph is out of context. Delete.

-        Introduction. In general, the authors must ensure they give enough background for their experiments.

-        Section 2.2. How did the authors select these germination conditions? Are those the best conditions for all the legumes investigated?

-        Table 1. What about the characteristic of the other legumes?

-        Section 2.3. What is the rationale for using these conditions for extracting legume phytochemicals?

-        Section 2.4. Why did the authors select Alcalase and Flavorzime as enzymes for their hydrolysis process? Why these conditions?

-        Line 148. Revise. There is no uptake of free radicals. “ABTS radical scavenging assay”

-        Why did the authors choose absorbances different from the common ones in ABTS and DPPH assays (620 and 493 nm vs. the standard 734 and 510-520 nm)? Why did they use gallic acid as standard in the ABTS assay instead of Trolox?

-        Section 2.7. How did the authors know the concentrations they were using were non-cytotoxic? Why did they use 2 H2O2 concentrations? Why those? Why did they use paraquat for protein isolates and H2O2 for ethanolic extracts? Why did the authors evaluate antioxidant capacity using MTT. They are instead evaluating cytoprotective effects. Change it. What were the concentrations assayed?

-        Line 208. Define GST and QR the first time used.

-        Why did the authors not characterize the composition of protein hydrolysates?

-        Section 2.10. Better explain combination assays.

-        The methods section does not follow the same other as the results.

-        Section 3.1. Errors are not needed in the text if included in the table.

-        All tables are pasted as figures. Revise it.

-        Why did the authors use different antioxidant assays for ethanolic extracts and protein hydrolysates? Why were some legumes just analyzed as raw flour?

-        Why “in vitro antioxidant activity” (cytoprotective effects) were analyzed in just five (raw and germinated samples)?

-        The chemical composition should be moved to the beginning of the results section. Did the authors quantify any of the identified compounds?

-        Why most of the IC50 in Table 10 are not defined? Can the authors provide the whole set of results at different concentrations in their supplementary material to complement these results?

-        Why are most of the results in sections 3.5 and 3.6 not shown?

-        The mansucript lacks any coherence. Most experiments were just performed with part of the samples. There is no logical order in the presentation of results. The authors should rewrite the whole section, reorganize tables and figures, and include all the information (even if those were negative results).

-        My biggest concern in this manuscript, even if the authors can reorganize and rewrite it, is the absence of physiological significance in the experimental design. The authors are assaying extracts and protein hydrolysates that may never reach the colon, sometimes uncharacterized. These compounds might be transformed and absorbed during gastrointestinal digestion, and only residual components, potentially different from those tested, may reach the colon.

Author Response

Antioxidant and chemopreventive activity from legumes of commercial interest in colorrectal cancer (antioxidants-1957640)

Reviewer: 1

Comment 1. Title. It should be revised to include “germination”, “extracts”, and “protein hydrolysates”. The authors are not investigating the effects of whole seeds but isolated compounds from them..

Answer: Thank you very much for your appreciation. We have improved the title following the reviewer advice to be more specific.

Comment 2. Abstract. The authors must rewrite it and ensure it follows a logical order (objective > methods > results > conclusions). Make sure you define abbreviations if used.

Answer: Thank you very much for your comment. Abstract has been modified following the logical order.

Comment 3. Lines 55-57. Rewrite.

Answer: Thank you, we appreciate your comment. We have rewritten the sentence.

Comment 4. Lines 60-64. Rewrite.

Answer: Answer: Thank you, I appreciate your comment. We have rewritten the sentence.

Comment 5.  Lines 69-76. This paragraph is out of context. Delete.

Answer: We have deleted the paragraph.

Comment 6.   Introduction. In general, the authors must ensure they give enough background for their experiments.

Answer: We appreciate your comment. We have improved the introduction giving more background.

Comment 7.  Section 2.2. How did the authors select these germination conditions? Are those the best conditions for all the legumes investigated?

Answer: The germinating conditions were selected based on previous experience of the research group (Urbano et al. 2005) that observed greater nutritional value resulting from either short- or mid-term germination lengths (2-4 days) either in the presence or absence of light. Such sprouting conditions resulted in foodstuffs with improved functional properties that exhibited significant benefits in the dietary treatment of obesity and metabolic syndrome (Kapravelou et al., 2017) using different legume species. Therefore, similar standardized conditions were used in the seeds chosen for the present study. The two references for germination methodology have been included in the Materials and Methods section 2.2.

Comment 8.  Table 1. What about the characteristic of the other legumes?

Answer: The effects of germination on the antioxidant and antitumor capacity of legumes were studied on faba beans due to the specific characteristics of the different seed varieties (low- or high-tannin content, presence or absence of vicine/convicine) that allowed for a more complete experimental design on that particular legume specie. That is the reason why Table 1 was specifically focused on faba bean characteristics and experimental design. Regarding the other legumes, only one variety of the legume seed was chosen and no germination treatment implemented. Therefore, the authors did not consider that a detailed explanation of that design was required on Table 1. Nevertheless, if the reviewer considers it should be necessary, we can include them in that table 1 (i.e. Table 1b).

Comment 9.   Section 2.3. What is the rationale for using these conditions for extracting legume phytochemicals?

Answer: Extracting conditions were selected to specifically obtain bioactive compounds of polyphenolic nature, avoiding the extraction of protein components that are usually denatured by such processing conditions. In addition, ethanol was chosen due to its lower toxicity compared to other organic solvents usually described in the literature such as methanol, acetone or DMSO. If the reviewer considers it necessary, the justification paragraph can be included in the materials and methods section 2.3.

Comment 10.    Section 2.4. Why did the authors select Alcalase and Flavourzime as enzymes for their hydrolysis process? Why these conditions?

Answer: We have selected sequential protein hydrolysis by Alcalase (endo-protease) and Flavourzyme (exo-protease) with different catalytic activities based on the protocol by Villanueva et (1999) with small modifications by Kapravelou et al (2013) to obtain a high-quality protein hydrolysate with efficient degree of hydrolysis, improved yield of water-soluble bioactive compounds and reduced bitterness. If the reviewer considers it necessary, the justification paragraph can be included in the materials and methods section 2.4.

Garyfallia Kapravelou, Rosario Martínez, Ana M. Andrade, Cristina Sánchez, Carlos López Chaves, María López-Jurado, Pilar Aranda, Samuel Cantarero, Francisco Arrebola, Eduardo Fernández-Segura, Milagros Galisteo, Jesús M. Porres, Health promoting effects of Lupin (Lupinus albus var. multolupa) protein hydrolyzate and insoluble fiber in a diet-induced animal experimental model of hypercholesterolemia, Food Research International, Volume 54, Issue 2, 2013, Pages 1471-1481.

Villanueva A, Vioque J, Sánchez-Vioque R, Clemente A, Bautista J, Millán F. Production of an extensive protein hydrolysate by sequential hydrolysis with endo- and exo-proteases. Grasas y Aceites, 50, 1999, 472-476.

Comment 11.    Line 148. Revise. There is no uptake of free radicals. “ABTS radical scavenging assay”

Answer: Thank you very much for your comment. We have changed it.

Comment 12.    Why did the authors choose absorbances different from the common ones in ABTS and DPPH assays (620 and 493 nm vs. the standard 734 and 510-520 nm)? Why did they use gallic acid as standard in the ABTS assay instead of Trolox?

Answer: Upon developing and implementing DPPH and ABTS methodologies on a microplate reader scale, the authors chose the nearest absorbance values of the filters available from the microplate reader that could be match those found in the literature. Since ABTS assay was performed in aqueous conditions, gallic acid gave a more stable reading than TROLOX and that is the reason why we selected that substance as our standard for the assay.

Comment 13.    Section 2.7. How did the authors know the concentrations they were using were non-cytotoxic? Why did they use 2 H2O2 concentrations? Why those? Why did they use paraquat for protein isolates and H2O2 for ethanolic extracts? Why did the authors evaluate antioxidant capacity using MTT. They are instead evaluating cytoprotective effects. Change it. What were the concentrations assayed?

Answer: Thank you very much for your comment.

  • Prior to carrying out this experiment, the functional extracts were tested in cell cultures to determine the IC50 value (Table 10). In this study, it was possible to corroborate the dose at which cell proliferation was not inhibited.
  • We used two doses of H2O2 to corroborate the efficacy of the extracts. One dose corresponds to a 50% inhibition by H2O2 and the other to a 60% inhibition. We have added this detail in line 209.
  • Paraquat was used specifically for the hydrolysates because H2O2 interfered with the proteins in this functional extract and the assay could not be performed correctly. However, with the ethanolic extracts, H2O2 could be used without problem.
  • This test consists of determining the antioxidant capacity in cell cultures in vitro. For this, the value of Ic50 and IC60 of H2O2 (pro-oxidant agent) will be determined. If, in the presence of functional extracts, cell death, due to H2O2, decreases, then the extracts are acting as antioxidants against said prooxidant agent that induces cell death. That is why we determine it by MTT, to determine cell viability (%PR) described in line 210.

-For this test, the cells were incubated with the different functional extracts at the first dose that did not produce cell growth inhibition, the doses being specific for each one.

Comment 14.    Line 208. Define GST and QR the first time used.

Answer: Thank you very much for your comment. We have changed it and now, the first time used is in line 262 (GST) and in line 275 (QR).

Comment 15.    Why did the authors not characterize the composition of protein hydrolysates?

Answer: Due to technological aspects. The equipment needed to analyze the peptide composition of our protein hydrolyzates was not available at the University of Granada. In addition, the matrix of protein hydrolyzates was too complex and gave some technical problems with the columns of UPLC-MS. The authors decided to include only data from the ethanolic extracts that were resolved without any technical problem by the equipment available and the experimental analytical conditions used.

Comment 16.    Line 208. Define GST and QR the first time used.

Answer: Thank you very much for your comment. We have changed it and now, the first time used is in line 262 (GST) and in line 275 (QR).

Comment 17.    Section 2.10. Better explain combination assays.

Answer: : Thank you very much for your comment. We have explained the combination assay with more detail.

Comment 18.    The methods section does not follow the same other as the results.

Answer: : Thank you very much for your comment. We have changed the order of the methods .

Comment 19.    Section 3.1. Errors are not needed in the text if included in the table.

Answer: Thank you very much for your comment. We have removed the errors in the text.

 Comment 20.    All tables are pasted as figures. Revise it.

Answer: Thank you very much for your appreciation. Indeed, the tables have been added as an image due to their size. Originally, we have them in word format as editable tables, so when the proof was edited, tables will be uploaded independently so that they can be added with the correct format.

Comment 21.     Why did the authors use different antioxidant assays for ethanolic extracts and protein hydrolysates? Why were some legumes just analyzed as raw flour?

Answer: Some of the analytical techniques implemented for the assessment of antioxidant capacity were suitable for both the ethanolic extracts and protein hydrolyzates (total polyphenol content, Fe-reducing capacity). Conversely, some techniques like the lipid peroxidation inhibition capacity were not compatible with the protein hydrolyzates (in fact, they exhibited pro-oxidant effect), but worked fine with ethanol-evaporated ethanolic extracts. As for DPPH and ABTS, they shared a similar principle but were carried out in methanol or aqueous media. Therefore, we decided to optimize the assay conditions of each methodology to ethanolic extracts or water-soluble protein hydrolyzates, respectively.

If the reviewer considers it necessary, the justification paragraph can be included in the materials and methods or results section.

The effects of germination on the antioxidant and antitumor capacity of legumes were studied on faba bean due to the specific characteristics of the different seed varieties (low or high-tannin content, presence or absence of vicine/convicine) that allowed for a more complete experimental design on that particular legume specie. For the other legumes, only one variety of the legume seed was chosen and no germination treatment implemented.

 Comment 22.    All tables are pasted as figures. Revise it.

Answer: Thank you very much for your appreciation. Indeed, the tables have been added as an image due to their size. Originally, we have them in word format as editable tables, so when the proof was edited, tables will be uploaded independently so that they can be added with the correct format.

 Comment 23.     Why “in vitro antioxidant activity” (cytoprotective effects) were analyzed in just five (raw and germinated samples)?

Answer: The in vitro antioxidant activity of all the samples was tested and presented in tables 8 and 9. The effects of germination on the antioxidant and antitumor capacity of legumes were studied on faba bean due to the specific characteristics of the different seed varieties (low or high-tannin content, presence or absence of vicine/convicine) that allowed for a more complete experimental design on that particular legume specie. For the other legumes, only one variety of the legume seed was chosen and no germination treatment implemented. To clarify these differences in experimental design, the results of faba beans and those of the rest of legumes assayed were presented separately.

 Comment 24.    The chemical composition should be moved to the beginning of the results section. Did the authors quantify any of the identified compounds?

Answer: Thank you very much for your appreciation. We have moved it to the beginning, now is the section 3.2. The bioactive compounds identified have not been quantified. In subsequent studies, it is intended to quantify them and even isolate them to see the antioxidant and antitumor activity they have independently.

 Comment 25.    Why most of the IC50 in Table 10 are not defined? Can the authors provide the whole set of results at different concentrations in their supplementary material to complement these results?

Answer: Thank you very much for the comments.  This table shows the IC50 values ​​of those functional extracts that had antitumor activity. The values ​​that are not shown are because they did not show antitumor activity. Likewise, a supplementary figure has been incorporated with the graphs of all the functional extracts (Figure S1 and S2).

Comment 26.    Why are most of the results in sections 3.5 and 3.6 not shown?

Answer: Thank you very much for your appreciation. The results have not been shown because they have not shown synergy in comparison with 4-Fu. And only those extracts have been tested where inhibition was observed in the T84 cell line, which did not show synergistic activity. Likewise, we have added a supplementary figure with those results (Figure S3). In the case of section 3.6, the study was only carried out with the selected extracts since they were the ones that obtained the best results in terms of antitumor and antioxidant activity.

Comment 27.    Why are most of the results in sections 3.5 and 3.6 not shown?

Answer: Thank you very much for your appreciation. The results have not been shown because they have not shown synergy in comparison with 4-Fu. And only those extracts have been tested where inhibition was observed in the T84 cell line, which did not show synergistic activity. Likewise, we have added a supplementary figure with those results (Figure S3). In the case of section 3.6, the study was only carried out with the selected extracts since they were the ones that obtained the best results in terms of antitumor and antioxidant activity.

Comment 28.    The mansucript lacks any coherence. Most experiments were just performed with part of the samples. There is no logical order in the presentation of results. The authors should rewrite the whole section, reorganize tables and figures, and include all the information (even if those were negative results).

Answer: Thank you very much for your appreciation. We have made the modifications that the referee have suggested to us so that it gains coherence and the results are better presented.

Comment 29.    My biggest concern in this manuscript, even if the authors can reorganize and rewrite it, is the absence of physiological significance in the experimental design. The authors are assaying extracts and protein hydrolysates that may never reach the colon, sometimes uncharacterized. These compounds might be transformed and absorbed during gastrointestinal digestion, and only residual components, potentially different from those tested, may reach the colon.

Answer: The results presented in this article have been obtained in the context of a research project aimed to develop new functional ingredients with high antioxidant capacity that may be used in the treatment of colon cancer and metabolic syndrome. The first step was to carry out a general screening in several legume seeds to assay the antioxidant capacity of two different functional ingredients. First, an ethanolic extract rich in polyphenolic bioactive compounds, and also a protein hydrolysate with bioactive peptides and water-soluble bioactive components. The following step has been to characterize the composition of ethanolic extracts and test the molecular mechanisms involved in the anti-tumor activity. All these studies have provided very valuable information to select the extracts with greater activity to prepare functional ingredients and nutraceuticals. These results comprise extensive information that is presented along 11 tables and 2 figures. In a second stage of the research project, the in vitro digestibility assays will be done to assess the potential absorption of bioactive compounds and provide further information on the effectiveness of the products developed.    

Reviewer 2 Report

The manuscript reported the properties of ethanolic extraction and protein hydrolysate derived from seven varieties of Vicia faba species, and other legume species as antioxidant, anti-proliferative and anti-migration agents in colorectal cancer cell lines. These results seem to be interesting but there are some major concerns in the manuscript that need to be addressed. 

Major concerns

-        Title should be revised to “Antioxidant and chemopreventive activity from legumes of commercial interest in colorectal cancer cell lines”

-        Why the author compares the different between raw and germinated for the varieties of V. faba but not other legume species? The discussion should be added.

-        The impact of protein hydrolysate from legumes on health benefits should be added to introduction section.

-        Table1: Since there is no report of tannin, vicine and convicine contents of the samples, how can the samples be classified as low/high tannins or low/high vicine-convicine?

-        L120: Why did the ethanolic extraction need to be done under a nitrogen atmosphere? The reference(s) for the extraction process and protein hydrolysate should be added.

-        More detail should be described for TBARs assay. How did the author obtain rat brain homogenate for experiment? How to generate lipid peroxidation in rat brain homogenate?

-        Why was HT-29 colon cancer cell line used for antioxidant and detoxifying enzyme induction assay? The authors should explain why the H2O2 was used as an oxidizing agent for testing antioxidant activity of ethanolic extract, whereas, for protein hydrolysates, paraquat was used instead of H2O2 to induce oxidation. Can the in vitro antioxidant activity be reported only for one concentration of H2O2? Any point of discussion that the authors would like to make?

-        How did the authors obtain “non-cytotoxic doses” of ethanolic extracts and protein hydrolysates for antioxidant, detoxifying enzyme induction, and cell migration assay?

-        For antioxidant activity, why the lipid peroxidation activity and DPPH scavenging activity were only analyzed for ethanolic extract whereas protein hydrolysate was studied only for ABTS scavenging activity?

-        The expression of data for antioxidant activity in cell lines as mUAA/mg is confusing and needs to be revised.

-        Why the different concentrations of ethanolic extract (40 µg/mL) and protein hydrolysate (25 µg/mL) were used for the induction of detoxifying enzyme activity assay?

-        For antiproliferative activity, please discuss the effect of testing compounds on the different cell types of colorectal cancer.

-        Spelling and grammatical errors need to be corrected.

Minor concerns

-        Conclusion of the abstract should be related to the findings of the study.

-        L91: glutathione

-        L125: “To determine the concentration of ethanolic extracts” Concentration of?

-        L148: “Uptake of free radicals (ABTS)” should be changed to “ABTS radical scavenging assay”

-        L179: Please add the full term of UUA

-        L187, 194:  H202 to H2O2

-        L207: on ice

-        L211: inactivation

-        L236: “Chromatograhic studies” should be changed to “Phytochemical analysis using liquid chromatography - mass spectrometry (LC-MS)”

-        L262: Cell survival (%) = [(Treated cells OD – blank)/(Control OD – blank)] × 100

-        L264: effect,

-        L307: polyphenols.

Author Response

Antioxidant and chemopreventive activity from legumes of commercial interest in colorrectal cancer (antioxidants-1957640)

Reviewer: 2

Comment 1.    Title should be revised to “Antioxidant and chemopreventive activity from legumes of commercial interest in colorectal cancer cell lines”

Answer: Thank you very much for your appreciation. We have improved the title to be more descriptive.

Comment 2.    Why the author compares the different between raw and germinated for the varieties of V. faba but not other legume species? The discussion should be added.

Answer: Thank you very much for your appreciation. We compared the difference between raw and germinated for the V. faba varieties because of the other species no germinated seeds were provided.

Comment 3.       The impact of protein hydrolysate from legumes on health benefits should be added to introduction section.

Answer: Thank you for your contribution. We have added new information about that in the introduction.

Comment 4.    Why the author compares the different between raw and germinated for the varieties of V. faba but not other legume species? The discussion should be added.

Answer: The effects of germination on the antioxidant and antitumor capacity of legumes were studied only on faba bean due to the specific characteristics of the different seed varieties (low or high-tannin content, presence or absence of vicine/convicine) that allowed for a more complete experimental design on that particular legume specie. For the other legumes, only one variety of the legume seed was chosen and no germination treatment implemented.

Comment 5.    L120: Why did the ethanolic extraction need to be done under a nitrogen atmosphere? The reference(s) for the extraction process and protein hydrolysate should be added.

Answer: The methodology employed to obtain the ethanolic extracts was designed to isolate polyphenolic substances. From the bibliographic review carried out to design the isolation process, the authors concluded that establishing a Nitrogen atmosphere was beneficial to avoid the oxidation of bioactive compounds. If the reviewer considers it necessary, the justification paragraph can be included in the materials and methods section

The references for ethanolic extraction and protein hydrolysis have been included in the Materials and Methods section as suggested by the reviewer.

Page 3, Line 127. Legume seed flours were used to obtain the ethanolic extracts according to Martinez et al. [2016] and Mesas et al. [2021].

Rosario Martínez, Garyfallia Kapravelou, Jesús M. Porres, Adela M. Melesio, Leticia Heras, Samuel Cantarero, Fiona M. Gribble, Helen Parker, Pilar Aranda, María López-Jurado, Medicago sativa L., a functional food to relieve hypertension and metabolic disorders in a spontaneously hypertensive rat model, Journal of Functional Foods, Volume 26, 2016, Pages 470-484, ISSN 1756-4646, https://doi.org/10.1016/j.jff.2016.08.013.

Mesas, C., Martínez, R., Ortíz, R., Galisteo, M., López-Jurado, M., Cabeza, L., Perazzoli, G., Melguizo, C., Porres, J. M., & Prados, J. (2021). Antitumor Effect of the Ethanolic Extract from Seeds of Euphorbia lathyris in Colorectal Cancer. Nutrients, 13(2), 566. https://doi.org/10.3390/nu13020566

Page 4, Line 139. Legume protein hydrolysates were prepared by a simultaneous process of alkaline water extraction and hydrolysis with proteases as described by Kapravelou et al [2015].

Comment 6. More detail should be described for TBARs assay. How did the author obtain rat brain homogenate for experiment? How to generate lipid peroxidation in rat brain homogenate?

Answer: As suggested by the reviewer, more details have been included for the TBARs assay in the materials and methods section. Page 5, Line 177. “The thiobarbituric acid reactive substances (TBARS) assay measures the capacity to inhibit lipid peroxidation occurring in a rat brain homogenate by ethanolic extracts. The assay comprises the reaction of malondialdehyde (MDA) with thiobarbituric acid (TBA) as described by Ohkawa et al.[23] . Rat brain homogenates were prepared according to Singh et al. [24] and Oboh et al. [25]. Brain tissue was homogenized (1:10 w/v) in cold KCl (1.15%)/Triton X-100 (0.1%). The homogenate was centrifuged at 2000×g, at 4 °C for 5 min and the clear supernatant collected and stored at −20 °C for lipid peroxidation assays. Once the extracts were prepared, 150 µL of the appropriately diluted ethanolic extract (ethanol evaporated) were mixed with 150 µL of brain homogenate, 1250 µL of KCl (1.15%), 100 µL of 5 FeCl3 (5mM), and 100 µL of H2O2 (1 mM). The mixture was incubated at 37 °C for 60 min, after which 1750 µL of the oxidation reaction were mixed with 1500 µL of  HCl (0.25M)/ TCA (15%)/DETAPAC (1.34 mM)/BHT (0.5%), 300 µL of SDS (8.1%), and 300 µL of TBA (3%). The mixture was incubated at 75 °C for 60 min, after which samples were left to cool and then centrifuged for 30 min at 2500×g. The supernatant was collected and absorbance was measured at 532 nm to detect TBARS formation. The percentage of inhibition was calculated as follows: % inhibition = 100 − [100 × (A1/A0)], where A0 is the absorbance of the control oxidative reaction prepared with ethanol-evaporated extracting solution and A1 is the absorbance of the oxidative reaction in the presence of ethanol-evaporated extract. One unit of antioxidant capacity is defined as the amount of sample capable of inhibiting 50% TBARS formation compared to the control. Results were expressed as mUAA (milliunits of antioxidant activity) per mg of extract.

Comment 7. Why was HT-29 colon cancer cell line used for antioxidant and detoxifying enzyme induction assay? The authors should explain why the H2O2 was used as an oxidizing agent for testing antioxidant activity of ethanolic extract, whereas, for protein hydrolysates, paraquat was used instead of H2O2 to induce oxidation. Can the in vitro antioxidant activity be reported only for one concentration of H2O2? Any point of discussion that the authors would like to make?

Answer:

  • The HT29 line was used for this test due to its characteristics, being the cell line that is usually used by most authors.
  • Paraquat was used specifically for the hydrolysates because H2O2 interfered with the proteins in this functional extract and the assay could not be performed correctly. However, with the ethanolic extracts, H2O2 could be used without problem.
  • We used two doses of H2O2 to corroborate the efficacy of the extracts. One dose corresponds to a 50% inhibition by H2O2 and the other to a 60% inhibition. We have added this detail in the text

Comment 8.    How did the authors obtain “non-cytotoxic doses” of ethanolic extracts and protein hydrolysates for antioxidant, detoxifying enzyme induction, and cell migration assay?

Answer: Thank you very much for your appreciation. Prior to carrying out this experiment, the functional extracts were tested in cell cultures to determine the Ic50 value (Table 10). In this study, it was possible to corroborate the dose at which cell proliferation was not inhibited. These data have been added as supplementary material (Figure S1 and S2).

Comment 9. For antioxidant activity, why the lipid peroxidation activity and DPPH scavenging activity were only analyzed for ethanolic extract whereas protein hydrolysate was studied only for ABTS scavenging activity?

Answer: Due to the complex matrix of the protein hydrolyzates, and the reaction conditions using FeCl3 and H2O2. The lipid-peroxidation inhibition assay did not show any antioxidant activity for those functional extracts, although it gave positive values in the ethanol-extracted ethanolic extracts that were included in the results section.

The assay conditions used for DPPH scavenging assay involve the use of organic solvents such as methanol that may react with certain components of the aqueous-extracted protein hydrolyzates and thus interfere with the assay. That is the reason why DPPH assay was only used for ethanol extracts, whereas a scavenging activity assay with a water-soluble ABTS reagent was performed in protein hydrolyzates.

Comment 10.    The expression of data for antioxidant activity in cell lines as mUAA/mg is confusing and needs to be revised.

Answer: Thank you very much for your appreciation. The results of this assay were expressed as antioxidant activity units (AAU), which is defined as the value of 10 percentage units (10%) of recovery of cell viability with respect to the corresponding control treated with hydrogen peroxide (H2O2). We have changed it as AAU in the text.

Comment 11.    Why the different concentrations of ethanolic extract (40 µg/mL) and protein hydrolysate (25 µg/mL) were used for the induction of detoxifying enzyme activity assay?

Answer: We first tested various concentrations of the ethanolic and hydrolyzed extracts and represented the concentrations where an induction on these enzymes was observed the effect of the protein hydrolysates at a lower concentration could be explained as Cai et al. 2017 due to the presence of bioactive peptides that enhance the activities of certain ethanolic compounds for example certain acidic amino acids such as Glu and Asp contributed to the antioxidant activities due to the presence of excess electrons that could be donated during the interaction with free radicals, in addition, peptides of certain proteins, produce an increase in hydrophobicity which would enhance their interaction with lipid targets or the entry of the peptides together with compounds from the ethanolic extract into the target organs through hydrophobic associations, thus enhancing their detoxifying effects.

Cai, X., Yan, A., Fu, N., & Wang, S. (2017). In Vitro Antioxidant Activities of Enzymatic Hydrolysate from Schizochytrium sp. and Its Hepatoprotective Effects on Acute Alcohol-Induced Liver Injury In Vivo. Marine drugs, 15(4), 115. https://doi.org/10.3390/md15040115

Comment 12.    For antiproliferative activity, please discuss the effect of testing compounds on the different cell types of colorectal cancer.

Answer: Several cell lines were measured to corroborate the antitumor activity that some of the varieties of legumes tested presented. Colon cancer lines T84 and SW240 were used, as well as a chemotherapy-resistant colon cancer line HCT-15. Similarly, a non-tumor cell line (CCD18) was used to determine selectivity for tumor cells.

Comment 13.    Spelling and grammatical errors need to be corrected.

Answer: Thank you very much for your appreciation. Grammatical errors have been reviewed and corrected.

Comment 14.    Conclusion of the abstract should be related to the findings of the study.

Answer: Thank you very much for your appreciation. We have improved the abstract following the referee’s advise.

Comment 15.    L91: glutathione

Answer: Thank you very much for your comment. It has been changed.

Comment 16.    L125: “To determine the concentration of ethanolic extracts” Concentration of?

Answer: It refers to knowing the concentration of the ethanolic extract and thus being able to carry out the in vitro tests with known concentrations of the functional extracts.

Comment 17.    L148: “Uptake of free radicals (ABTS)” should be changed to “ABTS radical scavenging assay”

Answer: Thank you very much for your comment. We have changed it.

Comment 18.     L179: Please add the full term of UUA

Answer: Thank you very much for your appreciation. We have unified the acronyms as mUAA (units of antioxidant activity)

Comment 19.     L187, 194:  H202 to H2O2

Answer: Thank you very much for your comment. We have changed it.

Comment 20.    L207: on ice

Answer: Thank you very much for your comment. We have changed it.

Comment 21.     L211: inactivation

Answer: Thank you very much for your comment. We have changed it.

Comment 22.    L236: “Chromatograhic studies” should be changed to “Phytochemical analysis using liquid chromatography - mass spectrometry (LC-MS)”

Answer: Thank you very much for your comment. We have changed it.

Comment 23.    L262: Cell survival (%) = [(Treated cells OD – blank)/(Control OD – blank)] × 100

Answer: Thank you very much for your comment. We have changed it.

Comment 24.     L264: effect,

Answer: Thank you very much for your comment. We have changed it.

Comment 25.    L307: polyphenols.

Answer: Thank you very much for your comment. We have changed it.

Reviewer 3 Report

Dear authors,

After the review process, I have several comments: you should include numerical data in the abstract; no comments in the introduction and discussion about microbiota that is essential in the management of colorectal cancer; you should include comments in section 4 about the correlations between microbiota bioactivity and bioavailability of functional compounds; also, you should add data related to the bioactive potential of functional products and bioavailability of phenolic compounds.

Best regards!

Author Response

Antioxidant and chemopreventive activity from legumes of commercial interest in colorrectal cancer (antioxidants-1957640)

Reviewer: 3

Comment 1.    After the review process, I have several comments: you should include numerical data in the abstract; no comments in the introduction and discussion about microbiota that is essential in the management of colorectal cancer; you should include comments in section 4 about the correlations between microbiota bioactivity and bioavailability of functional compounds; also, you should add data related to the bioactive potential of functional products and bioavailability of phenolic compounds.

Answer: Thank you for your comments. Following the referee’s advise we added numerical data in the abstract. Likewise, we have included information on the relationship between the microbiota and the development of colon cancer.

Round 2

Reviewer 1 Report

After carefully reading the manuscript’s revised version and the answers to all the reviewers’ comments, I still consider that this manuscript is not ready for publication.

I recommend that the authors separate the ethanolic extract and protein hydrolysates into two different manuscripts. Considering that most assays could not be run for both types of samples, there was no comparison available. Then, there is no significance in keeping them together.

I also recommend that the authors at least characterize the main characteristics of protein hydrolysates (SDS-PAGE electrophoresis, degree of hydrolysis, amino acid profiles, and, if viable, the peptide profile).

As the authors stated, This manuscript results from much work and includes extensive information. However, I feel that, in this case, more information is not better. I urge the authors to divide their manuscript, create a coherent story and explain the results obtained, rather than including many experiments without coherence.

Author Response

Answer: Thank you very much for your appreciation. In response to the referee's recommendation to improve the article and make it cohesive, a redistribution of the results has been carried out to focus on those obtained from protein hydrolysates.

We have made changes in the methodology, in the title of the article, as well as in the results and discussion section to focus the article on the results obtained with protein hydrolysates.

Protein hydrolysates are the functional extracts that showed the best results in terms of antioxidant activity as well as the only ones that showed antitumor activity. That is why we have discarded the results of the ethanolic extracts, maintaining those of the protein hydrolysate.

Regarding the characterization of the protein hydrolysate, specific studies will be carried out in future to determine specific peptides or amino acids that give them this powerful antioxidant and antitumor activity.

Reviewer 3 Report

Dear authors,

No other comments compared with the first review.

Best regards!

Author Response

Thank you very much

Round 3

Reviewer 1 Report

I still consider that the authors need to characterize their protein hydrolysates.

I suggested characterizing the main characteristics of protein hydrolysates (SDS-PAGE electrophoresis, degree of hydrolysis, amino acid profiles, and, if viable, the peptide profile).

There are many publications on the effects of legume peptides on colon cancer cells. I do not find innovative or robust results in this study. The antioxidant measure is based on ABTS, DPPH, cytoprotective effects, and the induction of GST and QR. However, colon cancer treatments aim to promote oxidative stress in cancer cells to trigger apoptosis. Then, I do not understand the rationale of those measures.

Besides, none of the stories presented by the authors was coherent (nor the peptide or the phenolic compounds one).

Author Response

Antioxidant and chemopreventive activity from legumes of commercial interest in colorrectal cancer (antioxidants-1957640)

Reviewer report

Comment 1. I still consider that the authors need to characterize their protein hydrolysates. I suggested characterizing the main characteristics of protein hydrolysates (SDS-PAGE electrophoresis, degree of hydrolysis, amino acid profiles, and, if viable, the peptide profile).

Answer:  Thank you very much for your comments. Currently we can not provide results about the characterization of the protein hydrolyzates. In fact,  the objective of this study was to know  the  extract antitumor activity against  colon cancer cells but not  the determination of the bioactive compounds.  In next studies,  and following referee suggestion, we will focus on fractioning the protein extract in order to characterize its components and determine which bioactive compound or compounds are responsible for the antitumor activity against colon cancer cells.

Comment 2. There are many publications on the effects of legume peptides on colon cancer cells. I do not find innovative or robust results in this study. The antioxidant measure is based on ABTS, DPPH, cytoprotective effects, and the induction of GST and QR. However, colon cancer treatments aim to promote oxidative stress in cancer cells to trigger apoptosis. Then, I do not understand the rationale of those measures.

Besides, none of the stories presented by the authors was coherent (nor the peptide or the phenolic compounds one).

Answer: Research for new treatments in cancer is highly relevant, which justifies the high number of articles published. These studies including  preventive natural compounds from plants show relevance in colon cancer due to  the high inicidence of this type  of tumor  in developed countries.

In relation to the reviewer's comment about antioxidant capacity and detoxifying enzymes of the extract, we should comment that many studies have shown that some bioactive molecules  are capable to make  antioxidant and antitumor effects at the same time thougth multitude of molecular mechanisms. The  cell death provoques by an antitumor agent does not necessarily have to induce by  oxidative stress (Costea et al., 2018).

The interest of our work is not only to find extracts that are effective against colon cancer but also that can prevent the appearance of this type of tumor. The study of the antitumor/anticancer capacity of a bioactive compound or a functional extract shows different aspects that should be studied. In any case, both antiproliferative and preventive effects  should be analyzed. We think that the approach and development of our results and discussion follows a logical and adequate line and is correctly adapted to the objectives of the work and our study hypothesis.

In any case, there are many studies that have analyzed this double effect. In fact,  Grigalius (Grigalius et al., 2017)  evaluated the antioxidant and antitumor activity of 17 flavones, finding a potent antitumor effect against lung cancer cell lines, which was also correlated with antioxidant activity. In addition, similar results were obtained in  plant extracts that have a strong antioxidant and antitumoral capacity. In fact, AI-Dabbagh (Al-Dabbagh et al., 2018) developed hydroalcoholic extracts from Trigonella foenum-graecum (fenugreek), Cassia acutifolia (Senna) and Rhazya stricta (Harmal), and demonstrated a strong antioxidant activity using the DPPH assay and a high antiproliferative activity against the human hepatoma cancer cell line (HepG2) which was attributed to its content in polyphenols and flavonoids.

  1. Costea, T., Hudiță, A., Ciolac, O. A., Gălățeanu, B., Ginghină, O., Costache, M., Ganea, C., & Mocanu, M. M. (2018). Chemoprevention of Colorectal Cancer by Dietary Compounds. International journal of molecular sciences, 19(12), 3787. https://doi.org/10.3390/ijms19123787
  2. Grigalius, I., & Petrikaite, V. (2017). Relationship between Antioxidant and Anticancer Activity of Trihydroxyflavones. Molecules (Basel, Switzerland), 22(12), 2169. https://doi.org/10.3390/molecules22122169
  3. Al-Dabbagh, B., Elhaty, I. A., Al Hrout, A., Al Sakkaf, R., El-Awady, R., Ashraf, S. S., & Amin, A. (2018). Antioxidant and anticancer activities of Trigonella foenum-graecum, Cassia acutifolia and Rhazya stricta. BMC complementary and alternative medicine, 18(1), 240. https://doi.org/10.1186/s12906-018-2285-7